# Cervical cancer screening rates before and after the Great East Japan Earthquake in the Miyagi Prefecture, Japan

**Yasuhiro Miki**[1], **Toru Tase**[2], **Hideki Tokunaga**[3], **Nobuo Yaegashi**[3], **Kiyoshi Ito**[1,4]*

**1** Department of Disaster Obstetrics and Gynecology, International Research Institute of Disaster Science, Tohoku University, Sendai, Japan, **2** Cancer Detection Center, Miyagi Cancer Society, Sendai, Japan, **3** Department of Obstetrics and Gynecology, Tohoku University Graduate School of Medicine, Sendai, Japan, **4** Disaster Medical Science Group, Core Research Cluster of Disaster Science, Tohoku University, Sendai, Japan

\* kito@med.tohoku.ac.jp

**Data Availability Statement:** All relevant data are within the manuscript and its Supporting Information files.

**Funding:** The authors received no specific funding for this work.

## Abstract

After disasters, issues pertaining to women's health such as irregular periods and bleeding are well surveyed. However, the management of women's health, especially changes in the rate of health checkups, has not been investigated. In the present study, we focused on the change in the cervical cancer screening rates (CCS-Rs) before and after the Great East Japan Earthquake in Miyagi Prefecture, Japan. The earthquake had a magnitude of 9.0, a profound disaster. We examined the CCS-R from 2009 to 2016 in 45 areas of the Miyagi Prefecture. Screening was completed using mobile vans. In the 4 areas impacted by the tsunami after the earthquake, a marked decrease in the CCS-R was observed in 2011 when the earthquake took place (more than a 3% decrease compared with that in the previous year). The CCS-Rs in these 4 regions remained lower in 2016 than in the previous year. In 2009–2016 except for 2014, CCS-Rs in coastal areas (9 areas) were significantly lower than those in the non-coastal areas (36 areas). A delay in seeking healthcare, also known as "patient's delay," is considered as one of the problems of cancer treatment in affected areas. It is possible that a decrease in the CCS-R may lead to low detection of advanced stages of cancer. Therefore, the establishment of a comprehensive medical system including medical screening after a disaster is important for the management of women's health.

## Introduction

The cervical cancer incidence in Japan is 14.7 (age-standardized rates per 100,000 population), which is higher than that in other developed countries such as United States (6.5), Australia (6.0), and South Korea (8.4) and similar to that in India (14.7) and the Philippines (14.9) [1]. Furthermore, vaccination coverage in Japan (under 1%) is significantly lower than that in other countries where cervical cancer vaccines are licensed [2,3]. Therefore, cervical cancer screening is considered crucial. However, the screening rate in Japan is considerably lower than that in Western and other countries [4]. It is reported that the screening rates in the

**Competing interests:** The authors have declared that no competing interests exist.

United States and Britain and those in France and South Korea are approximately 80% and 70%, respectively [4]. Furthermore, the screening rate in Japan (42.3% [age group: 20–69 years]) is lower than that in Australia (56%) [4–6]. An increase in the coverage of women receiving periodic cervical cancer screening is urgently needed in Japan. The rate of cervical cancer screening in the Miyagi Prefecture (51.7%; age group 20–69 years) was the second highest among the 47 prefectures in Japan in 2016 [6]. In the Miyagi Prefecture, the mobile van service (or mobile examination unit) offering cervical cancer screening serves rural areas where it is difficult to obtain an examination in a hospital or medical center [7,8]. There are two types of cancer screenings: population-based screening and opportunistic screening such as private and company medical check-ups. All of the mobile van screenings are population-based screenings.

On March 11, 2011, a magnitude 9.0 earthquake, termed the Great East Japan Earthquake, struck the east area of Japan. The abbreviation "3.11" has been used for this earthquake, mainly by the Japanese mass media as the earthquake occurred on March 11, 2011. After the Great East Japan Earthquake, higher-than-expected tsunami waves hit coastal zones of Tohoku and caused devastating damage. The economic loss was serious, and the damage caused by the earthquake was estimated to be approximately 16–25 trillion yen [9]. The tsunami-impacted area was predicted to require 10 years for reconstruction [10]. In the Miyagi Prefecture, a mobile van service offering cervical cancer screening covered the areas severely affected by the tsunami after the Great East Japan Earthquake. The screening resumed promptly to maintain the health of the residents in the disaster area. In many areas, cervical cancer screening was resumed in April of the year of the disaster, 2011. However, in coastal areas (L-2, L-3, T-3, U-1b in Fig 1), the re-initiation of screenings was delayed from July to December, 2011. The screening was resumed only in Oshika (L-7 in Fig 1) in February, 2012 following the earthquake. Unfortunately, the people affected by the disaster continue to be mentally and financially unstable and cannot afford to maintain their health [11,12,13].

It is well known that conflicts and disasters have a major impact on healthcare and result in delays in the diagnosis and treatment of patients with cancer [11,12]. Furthermore, the social isolation caused by a disaster may delay the opportunity to examine and treat patients with cancer [13]. A study at a city hospital in 2005 in New Orleans, affected by Hurricane Katrina, found that the mean time from last cytology (Pap test) to diagnosis of cervical cancer was significantly longer after the hurricane (7.7 years) than before the hurricane (4.2 years) (Abstract of the 2001 Western Association of Gynecologic Oncologists Annual Meeting, doi: 10.1016/j.ygyno.2011.07.082 [14]). In addition, the clinical stage of cervical cancer at diagnosis was significantly higher after the hurricane than before the hurricane [14]. Women's health hazards such as abnormal menstrual cycles, pelvic inflammation, and lower genital tract infections after disasters also have been reported in various countries [15–19]. Although the failure of health management is considered to lead to long-term health risks, the cervical cancer screening rate (CCS-R) after a disaster remains unexamined. In this study, we examined the trend of CCS-Rs from the Japan fiscal year (April 1 to March 31) 2009–2016 in 45 areas of the Miyagi Prefecture where cervical screenings were performed by the Miyagi Cancer Society (Fig 1, S1 Table).

## Materials and methods

CCS-Rs were obtained from the Annual Report in 2009–2016 and were compiled by the Miyagi Cancer Society (Sendai, Miyagi, Japan). The Miyagi Cancer Society was established as a public interest incorporated foundation and has formed a cancer screening and cancer registration business in Miyagi Prefecture, Japan. The annual report is also compiled as one of the

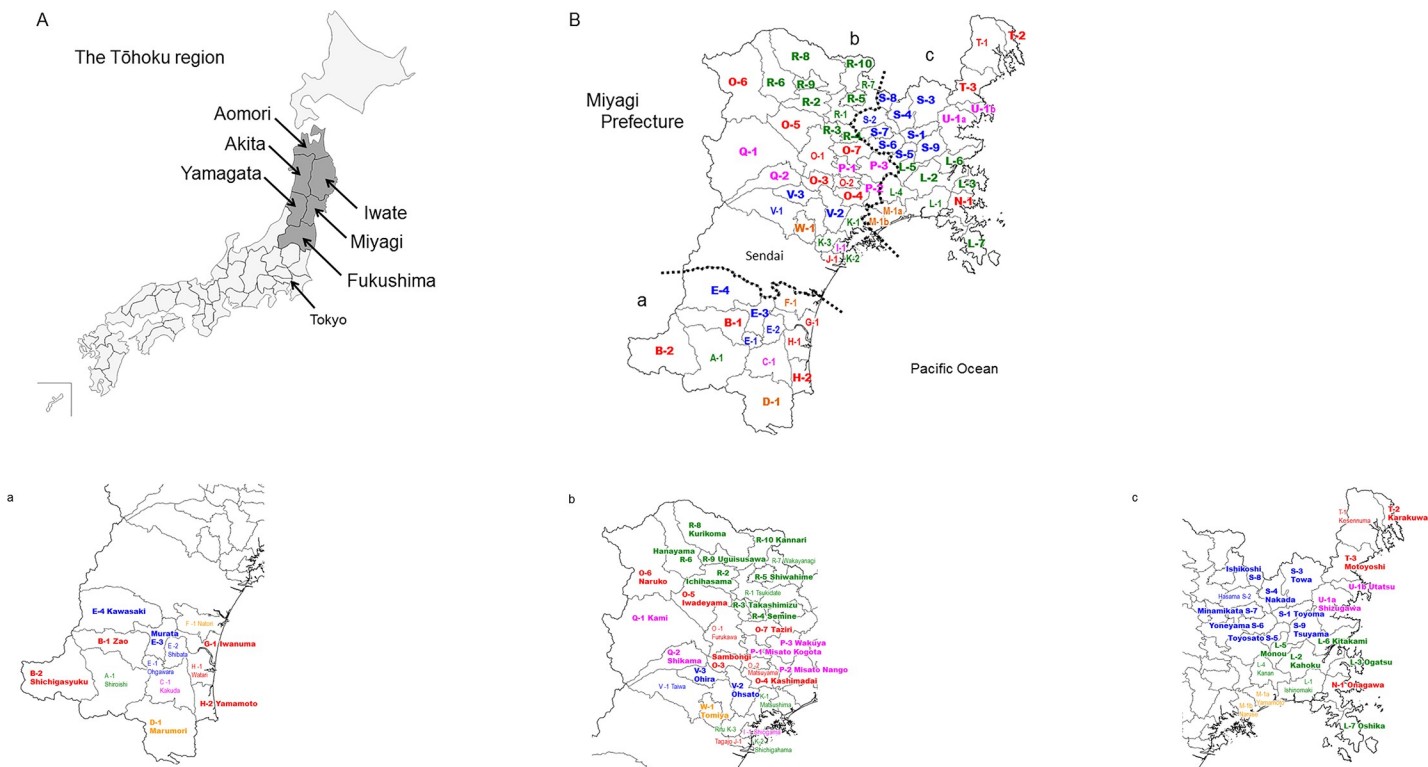

**Fig 1. The Tohoku region in Japan and the survey area in the Miyagi Prefecture.** (A) The Tohoku Region has six prefectures: Aomori Prefecture, Iwate Prefecture, Miyagi Prefecture, Akita Prefecture, Yamagata Prefecture, and Fukushima Prefecture. (B) The survey area in Miyagi. Enlarged views are presented for each of the areas a, b, and c. Also, see S1 Table for details. Bold style code numbers represent areas where cervical cancer screenings were performed by mobile vans.

functions of the Miyagi Cancer Society. The population of each region in each year was also extracted from this Annual Report.

The incidence of cervical cancer has increased in Japanese women in their mid-20s [20]. Therefore, Japanese cervical cancer screening guidelines recommend to begin screening at the age of 20 years [21]. The CCS-R was defined by the following equation: CCS-R = {(Number of women aged 20 years and older who underwent cervical cancer screening) ÷ (Total population of women aged 20 years and older)} × 100 (%). The difference in CCS-R for each year compared with the year before the Great East Japan Earthquake was calculated as follows: Δ [CCS-R in the year] = {CCS-R in the year (%)} − {CCS-R in the year before the Great East Japan Earthquake, 2010 (%)}; that is, Δ[CCS-R 2011] (%) = CCS-R in 2011 (%) − CCS-R in 2010 (%). The difference in population in each area was calculated similarly: Δ [Pop. in the year] = 100 (%) − {(Pop. in the year) ÷ (Pop. in the year before the earthquake, 2010) × 100 (%)}.

In the Miyagi Prefecture, cervical cancer screening was performed in the mobile van or the medical center/hospital. The medical exam statistics at the medical center or hospitals in the Miyagi Prefecture included a large number of sporadic outpatients, who complained of symptoms such as irregular bleeding. Therefore, the results of screening at medical facilities (23 areas) are shown in S1 Fig. The study area (45 areas) is shown in Fig 1 and S1 Table. In those regions, cervical cancer screening was provided by a mobile van. All of the data were processed using Microsoft Excel (Microsoft Corporation, Redmond, WA, USA). Mann–Whitney U test with StatView ver. 5.0 (SAS Institute Inc., San Francisco, CA, USA) was used for comparison between the groups. This research was approved by the ethics committees of the International

Research Institute of Disaster Science, Tohoku University (Sendai, Japan) and Miyagi Cancer Society (Sendai, Japan).

## Results

Differences in the CCS-R in each region from 2010 to 2016 are summarized in Table 1. Among the areas covered by the mobile van, the Δ[CCS-R 2011] in the 4 regions markedly decreased (over −3.0%) (Fig 2A): Ogatsu −5.2% (L-3), Onagawa −7.0% (N-1), Karakuwa −4.8% (T-2), and Shizugawa −4.1% (U-1a). These four coastal regions were affected by the tsunami. In Ogatsu (L-3), the CCS-Rs tended to recover the year after the earthquake (2013) and decreased approximately by 2.5% in 2013–2015 compared with the rate one year prior to the earthquake (2010). However, the Δ[CCS-R 2016] was −4.1% (Fig 2Ab). The population from 2011 (Δ[Pop.2011]) to 2016 (Δ[Pop.2016]) in Ogatsu (L-3) remained at −30.3% to −56.8% (Fig 2Aa). In Onagawa (N-1), where the decrease in the CCS-R was the highest in the year the earthquake occurred (Δ[CCS-R 2011], −7.0%), a high reduction ratio (−5.2 → −5.0 → −5.6 → −6.9 → −4.5) was observed even after the earthquake from (Δ[CCS-R 2012] to Δ[CCS-R 2016]) (Fig 2Ab). The Δ[Pop.2011] to Δ[Pop.2016] in Onagawa (N-1) remained at −20.6% to −34.8% (Fig 2Aa). In Karakuwa (T-2), although the CCS-R recovered to a decreased rate of less than 2.0% (Δ[CCS-R 2012]), it decreased by 3.0% from (Δ[CCS-R 2013] to Δ[CCS-R 2016]) (Fig 2Ab). The Δ[Pop.2011] to Δ[Pop.2016] in Karakuwa (T-2) remained at −4.0% to −11.1% (Fig 2Aa). In Shizugawa (U-1a), the CCS-R decreased by 3% in the year following the earthquake (Δ[CCS-R 2012]) but recovered to the level one year before the earthquake (2010) in 2013 after the merger with Utatsu (U-1b) and increased by more than 3.0% in 2015 and 2016 (Table 1). The Δ[Pop.2011] to Δ[Pop.2016] in Minamisanriku (U-1 = U-1a + U-1b) remained at −10.5% to −19.1% (Table 1).

In the tsunami-affected coastal areas other than the above four towns, the Δ[CCS-R 2011] was as follows (Fig 2B): Yamamoto−0.8% (H-2), Kahoku −1.1% (L-2), Kitakami +1.3% (L-6), Oshika −2.9% (L-7), and Motoyoshi −2.3% (T-3) (Fig 2Bab). The Δ [Pop. 2011] was as follows; Yamamoto −11.3% to −22.4% (H-2), Kahoku −3.5% to −7.0 (L-2), Kitakami −15.3% to −31.7% (L-6), Oshika −14.0% to −38.7% (L-7), and Motoyoshi −3.7% to −6.4% (T-3) (Fig 2Ba).

In non-coastal areas, the CCS-R declined by more than 3.0% compared with that of the year before the earthquake during the survey period in four places [Shichigashuku (B-2), Kahoku (L-2), Monou (L5), Tomiya (W-1)] (Fig 3A). In contrast, in the 4 non-coastal regions, the CCS-R increased by more than 3.0% during the survey period compared with that before the earthquake year [Kawasaki (E-4), Kami (Q-1), Shikama (Q-2), Tomiya (S-5)] (Fig 3B). No notable increase or decrease (±3.0%) was observed in other non-coastal areas (Fig 4).

When the areas examined were divided into coastal and non-coastal areas, the CSS-R was significantly lower in the coastal areas during 2011 (*p = 0.0046*), 2012 (*p = 0.0020*), 2013 (*p = 0.0247*), 2015 (*p = 0.0370*), and 2016 (*p = 0.0135*) (Fig 5). No significant difference was found in the 2-year comparison before the earthquake (2010–2009, *p = 0.2018*) and 2014 (2014–2010, *p = 0.6303*) (Fig 5).

## Discussion

The health of women, especially pregnant women, has been investigated in the areas affected by the Great East Japan Earthquake [22–24]; however, no research has been conducted with regard to cancer screening. In the present study, we first clarified the difference in the CCS-R of the Miyagi Prefecture before and after the Great East Japan Earthquake. In the present study, after the Great East Japan Earthquake, the CCS-R significantly decreased in the coastal areas compared with that in other areas. The coastal areas in the Miyagi Prefecture were

**Table 1. Differences in the cervical cancer screening rates between before and after the Great East Japan Earthquake.**

| Code No. | | | 2010–2009 | 2011–2010 | 2012–2010 | 2013–2010 | 2014–2010 | 2015–2010 | 2016–2010 |
|---|---|---|---|---|---|---|---|---|---|
| B | 1 | Zao Town | 1.5 | −0.8 | −1.8 | −1.0 | −1.8 | −0.6 | −1.7 |
| | 2 | Shichigasyuku Town | −1.4 | 0.0 | −1.7 | −1.7 | *−4.5* | *−3.3* | *−3.4* |
| D | 1 | Marumori Town | 0.7 | −0.2 | 0.9 | −0.1 | 0.1 | 1.8 | 0.5 |
| E | 3 | Murata Town | 0.4 | 0.5 | −0.4 | 0.5 | 0.7 | 0.8 | 1.4 |
| | 4 | Kawasaki Town | *6.4* | *3.8* | *4.5* | *3.2* | *4.7* | *7.3* | *6.2* |
| H | 2 | Yamamoto | 1.0 | −0.8 | −1.1 | −1.7 | −0.7 | −0.9 | −0.9 |
| L | 2 | Kahoku | 0.3 | −1.1 | −1.8 | −1.9 | −0.9 | −1.5 | *−3.9* |
| | 3 | Ogatsu | 0.2 | *−5.2* | *−4.0* | −2.4 | −2.6 | −2.4 | *−4.1* |
| | 5 | Monou | 0.3 | −0.6 | −2.3 | *−3.4* | −2.7 | *−3.5* | *−5.6* |
| | 6 | Kitakami | 0.3 | 1.3 | 0.0 | −1.0 | 0.6 | −2.2 | *−4.3* |
| | 7 | Oshika | −0.8 | −2.9 | −1.9 | −1.7 | 0.7 | *−3.6* | −1.8 |
| N | 1 | Onagawa Town | 0.8 | *−7.0* | *−5.2* | *−5.0* | *−5.6* | *−6.9* | *−4.5* |
| O | 3 | Sambongi | −0.6 | 1.7 | 1.7 | 1.7 | *−4.3* | 1.4 | 2.2 |
| | 4 | Kashimadai* | 1.4 | 0.5 | 1.4 | 1.1 | 1.3 | 0.8 | 2.0 |
| | 5 | Iwadeyama† | 0.6 | 1.3 | 1.6 | 1.3 | 1.8 | 2.0 | 2.8 |
| | 6 | Naruko | 0.2 | 1.1 | 0.8 | −0.9 | 0.2 | −1.4 | −0.1 |
| | 7 | Taziri | 2.1 | 0.9 | 0.5 | 0.6 | 1.2 | 0.3 | 2.0 |
| P | 1 | Misato Town Kogota | 0.2 | 0.1 | −1.0 | 0.0 | −0.8 | 2.6 | 2.1 |
| | 2 | Misato Town Nango | 0.5 | −1.9 | −2.4 | −1.8 | −2.2 | 0.5 | −1.3 |
| | 3 | Wakuya Town‡ | *3.9* | −1.2 | −2.9 | −1.5 | −1.2 | −1.9 | −0.9 |
| Q | 1 | Kami Town | −0.7 | 0.5 | 0.4 | 2.1 | 0.9 | 1.8 | *3.9* |
| | 2 | Shikama Town† | −1.4 | 1.4 | 2.3 | *3.2* | *4.7* | *4.7* | *4.7* |
| R | 2 | Ichihasama | 0.8 | −0.9 | −0.4 | −0.6 | −0.4 | −2.9 | 0.6 |
| | 3 | Takashimizu | 1.4 | 0.3 | −0.8 | 1.5 | 1.4 | 1.4 | 2.0 |
| | 4 | Semine | 1.3 | −1.0 | −1.3 | −2.5 | −0.3 | 1.1 | 0.5 |
| | 5 | Shiwahime | 1.3 | −0.5 | −1.3 | −2.1 | −0.8 | 0.4 | −0.4 |
| | 6 | Hanayama | 0.9 | −1.7 | −1.2 | 0.0 | −2.3 | 0.4 | −1.2 |
| | 8 | Kurikoma | 1.0 | 0.1 | −1.1 | −0.8 | −0.6 | −0.2 | −0.6 |
| | 9 | Uguisusawa | 1.0 | 0.1 | −0.2 | −1.0 | −1.1 | −0.8 | −0.5 |
| | 10 | Kannari | 0.3 | 0.3 | −0.3 | −0.4 | −0.2 | −1.4 | −0.1 |
| S | 1 | Toyoma | 0.4 | −0.4 | 0.2 | −0.2 | −0.5 | −0.1 | 0.1 |
| | 3 | Towa | −0.3 | 0.0 | 1.8 | 0.8 | 1.6 | 2.7 | 2.3 |
| | 4 | Nakada | 1.3 | −0.7 | 2.4 | 0.2 | 0.1 | 0.8 | −0.1 |
| | 5 | Toyosato | −0.7 | 1.0 | 2.3 | *3.0* | 2.6 | *4.2* | *3.3* |
| | 6 | Yoneyama | −0.5 | 1.5 | 2.0 | 2.3 | 1.3 | 1.1 | 0.8 |
| | 7 | Minamikata | 0.1 | −0.8 | −0.3 | −0.2 | −1.4 | −1.4 | −2.0 |
| | 8 | Ishikoshi | 1.4 | −1.6 | −0.2 | 0.0 | 0.2 | −1.2 | −0.1 |
| | 9 | Tsuyama | 1.5 | −1.9 | −0.8 | −0.5 | 0.3 | 1.6 | 2.3 |
| T | 2 | Karakuwa | −0.3 | *−4.8* | −1.9 | −2.7 | −2.5 | *−3.5* | −2.7 |
| | 3 | Motoyoshi | 1.9 | −2.3 | −1.3 | −1.1 | −1.8 | −2.6 | −2.6 |
| U | 1a | Shizugawa | 0.3 | *−4.1* | *−3.1* | 1.7 | 2.7 | *3.6* | *3.1* |
| | 1b | Utatsu | −0.4 | −2.7 | *−4.1* | – | – | – | – |
| V | 2 | Ohsato Town | −0.2 | −0.4 | −0.7 | 1.0 | 1.1 | 0.0 | 0.2 |
| | 3 | Ohira Village | 0.2 | −0.6 | 0.0 | −0.4 | −0.4 | −0.4 | 1.0 |
| W | 1 | Tomiya City | 1.4 | −0.3 | −2.6 | −1.2 | −2.0 | *−3.6* | −1.9 |

Areas covered by mobile vans were examined. Screening by the mobile vans *since 2015, †since 2013, and ‡since 2011; Bold italic, change of 3% or more; U-1, Shizugawa (U-1a) and Utatsu (U-1b) were integrated in 2013. The 2010 cervical cancer screening rate was compared with that of 2009.

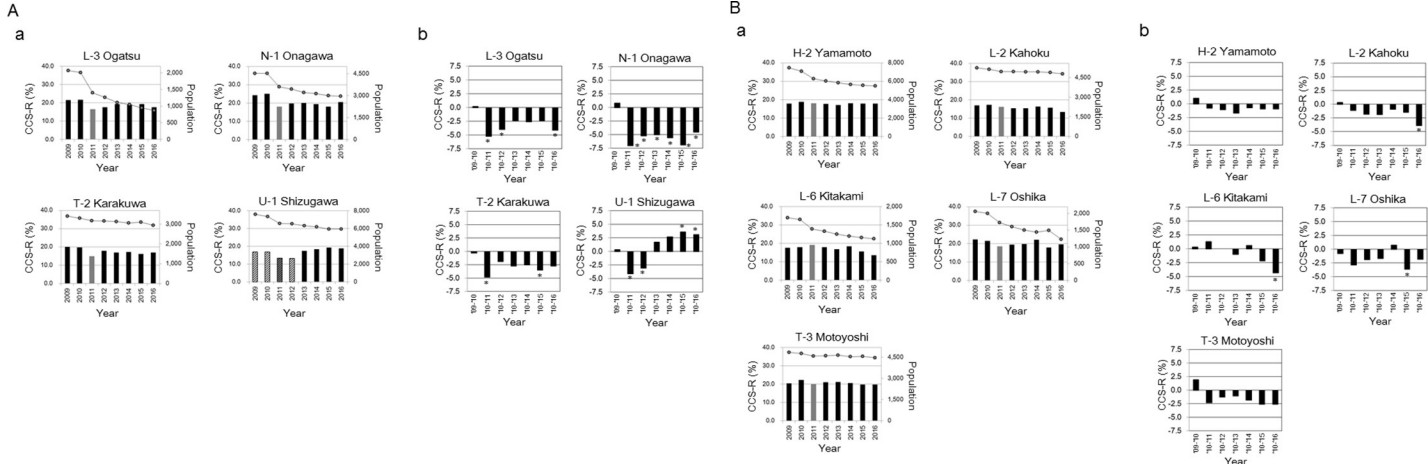

**Fig 2. Change in the CCS-R in coastal areas.** (A) a. Coastal areas showing a remarkable decrease in the cervical cancer screening rates (CCS-Rs) in the year following the Great East Japan Earthquake (over −3.0%). The bar graph shows the CCS-R (%), and the line graph shows the population (number of people). b. Differences in the CCS-R between before and after the Great East Japan Earthquake in each area demonstrated in "a". *, change of 3% or more. U-1 since Shizugawa (U-1a) and Utatsu (U-1b) were integrated in 2013; the data for 2009–2012 include a total of two regions. (B) a. Change in CCS-R in other coastal areas. The bar graph shows the CCS-R (%), and the line graph shows the population (number of people). b. Differences in the CCS-R between before and after the Great East Japan Earthquake in each area demonstrated in "a". *, change of 3% or more.

severely impacted by an unforeseen tsunami. Even within the screening areas, a drastic population decline was observed immediately after the earthquake. It is well known that the tsunami resulted in severe human and residential damage as a result of the Great East Japan Earthquake [25,26]. Therefore, coastal populations are considered to have decreased both due to death as well as migration as a consequence of the tsunami. After the Great East Japan Earthquake, the population also decreased in areas with a severe decrease in the CCS-R. Although the decrease in the CCS-R could have been attributed to the decline in the population, there were many areas in which there was no clear relationship between the decrease in population and the CCS-R. In the present study, we used the total population of each region for evaluation. We found that further studies that are corrected for the population of women aged over 20 years are required. The economic loss caused by the Great East Japan Earthquake in the Miyagi Prefecture was more severe in coastal areas than in inland areas (4.9 vs 1.6 trillion yen) in 2012

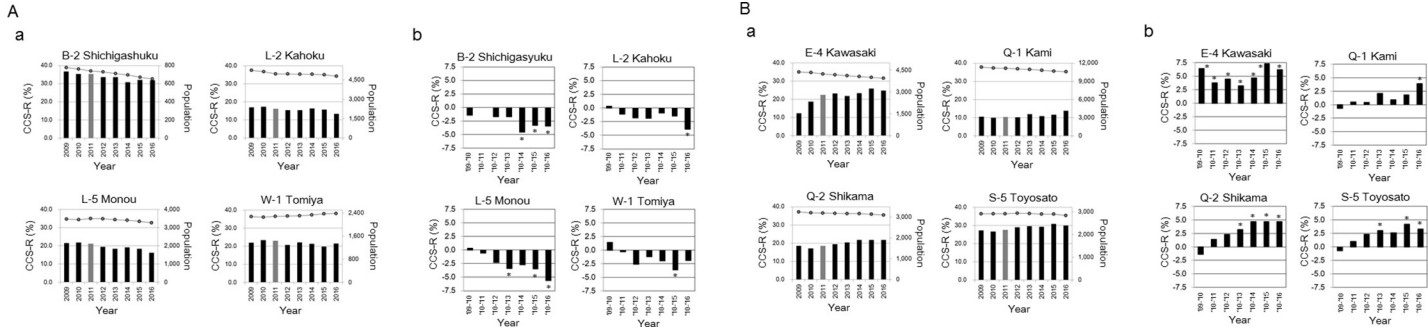

**Fig 3. Change in the cervical cancer screening rate (CCS-R) in non-coastal areas examined in this study.** (A) a. The CCS-R declined by more than 3.0% compared with that of the year before the earthquake (over −3.0%). The bar graph shows the CCS-R (%), and the line graph shows the population (number of people). b. Differences in the CCS-R between before and after the Great East Japan Earthquake in each area demonstrated in "a". *, change of 3% or more. (B) a. The CCS-R declined by more than 3.0% compared with that of the year before the earthquake (over +3.0%). The bar graph shows the CCS-R (%), and the line graph shows the population (number of people). b. Differences in the CCS-R between before and after the Great East Japan Earthquake in each area demonstrated in "a". *, change of 3% or more.

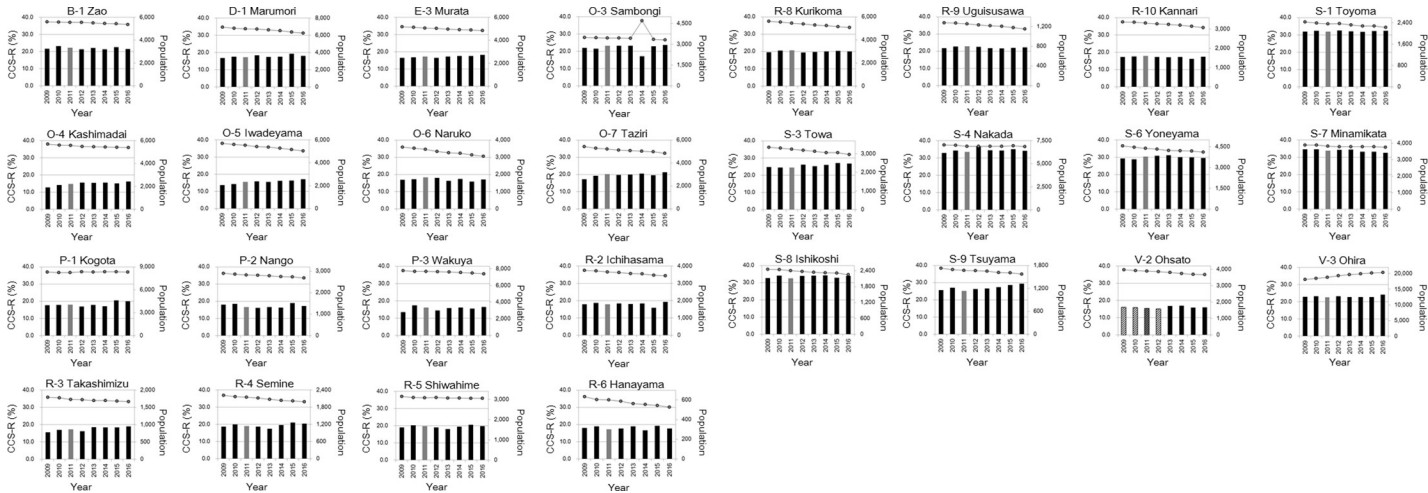

**Fig 4. Change in the cervical cancer screening rate (CCS-R) in other non-coastal areas examined in this study.** The bar graph shows the CCS-R (%), and the line graph shows the population (number of people).

[25]. In coastal areas, the damage to livelihoods and social infrastructure resulted in a loss of 2.0 trillion yen and the loss due to residential damage was 1.4 trillion yen. This was 2.4- and 36.2-times higher than that in inland areas, respectively [25]. Furthermore, reconstruction in the Miyagi Prefecture was reported to be markedly lower in coastal areas than in inland areas in 2012 to 2016 [26].

From the year after the Great East Japan Earthquake to 2016, there were some areas where the CCS-R recovered; however, other areas did not show such recovery. Furthermore, in many areas where cervical cancer screening had been conducted at medical facilities by the Miyagi Cancer Society, a decline in the CCS-R was observed in the year following the earthquake (S1 Fig). In an evidence review by CPSTF (USA), small media sources such as newspapers and educational videos as well as solicitation by telephone or letter are recommended as interventions to increase the CCS-R (https://www.thecommunityguide.org/content/task-force-findings-cancer-prevention-and-control; accessed 2020.1.17). In Japan, it was also reported that local government initiatives and education contributed to increasing the screening rate of young women [27]. However, a detailed investigation is required in areas where these screening rates have declined and awareness attempts by local government and other organizations may have been delayed. It is important for future post-disaster measures to clarify the details of the recruitment of cervical cancer screening programs for women in each area.

It is well known that conflicts and disasters have a major influence on healthcare and cause delays in diagnosis and treatment of patients with cancer [11]. In the Fukushima Prefecture, which was impacted by a triple disaster (an earthquake, a tsunami, and a nuclear accident), delays in symptom recognition to the first medical examination in patients with breast cancer have been reported [12]. The trend of risk due to patients' delay continues even following the disaster [12]. It is also reported that social isolation due to the triple disaster in Fukushima may have delayed opportunities for medical consultation and appropriate therapy among patients with colon cancer [13]. A survey of 366 patients with cervical cancer from urban hospitals in New Orleans, USA affected by Hurricane Katrina in 2005 was conducted [14]. The mean time from last cytology (Pap test) to diagnosis of cervical cancer was significantly longer after the hurricane (7.7 years) than before the hurricane (4.2 years) [14]. In addition, the cancer stage was significantly higher ($p < 0.01$) after the hurricane (From 2006 to 2010: stage I, 40

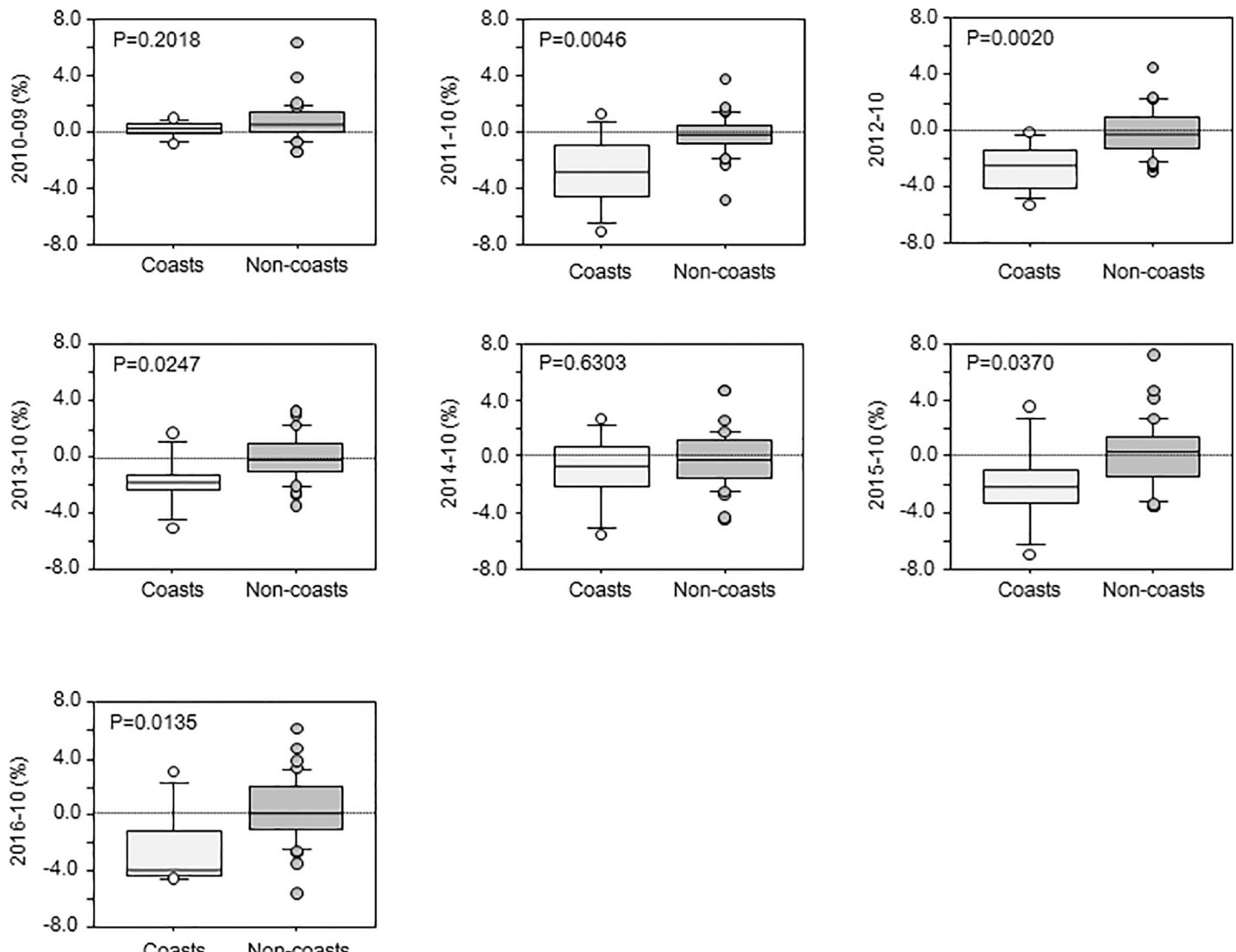

**Fig 5. Comparison of the change in the cervical cancer screening rate in coastal and non-coastal regions.** The bar graph shows the median value of differences in the CCS-R between before and after the Great East Japan Earthquake (%). Coast, nine coastal regions; Non-coasts, 36 non-coastal regions.

cases; stage II, 36 cases; stage III, 32 cases; stage IV, 32 cases.) than before the hurricane (From 2000 to 2005: stage I, 101 cases; stage II, 60 cases; stage III, 37 cases; stage IV, 28 cases) [14]. The five-year overall survival rates for cervical cancer stages I and II are known to be approximately 90% and 75%, respectively [20]. Early detection by periodic screening is critical in order to prevent death from cervical cancer [21]. If regular screening is skipped as a result of the earthquake, it may be detected later as advanced cancer. This situation was found in the area that was impacted by Hurricane Katrina [14]. These reports illustrate the importance of prompt reconstruction of healthcare systems, including screening, for early detection [14,28]. Compared to the USA CCS-R, the Japan CCS-R is too low [4,6]. Therefore, it would be difficult to perform the same analysis as that in US research due to the small number of comparative patients. Otherwise, in Japan, cervical cancer screening is expected to reduce cervical cancer deaths (recommendation grade B), with evidence levels of 2++ for conventional and 2+ for liquid-based cytology [21]. There are limitations to conducting a further detailed analysis

in the areas examined in this study; however, a reduction in CCS-Rs suggests that women's health will be significantly affected.

The trauma experienced during tsunamis remains a strong psychological stressor among disaster victims [29–33]. A survey after the Great East Japan Earthquake reported that difficulty settling back into daily living, the presence of preexisting illnesses, and the disruption of social networks were the most important risk factors for mental health disorders. These conditions did not improve even 2 years after the disaster [34]. Therefore, it is possible that women in the affected areas will continue to be exposed to long-term distress. Psychological stress is believed to exhibit immunosuppressive effects through the inhibition of several lymphocyte functions. Stress is known to be involved in tumorigenesis or tumor development by the inactivation of cytotoxic T cells and natural killer (NK) cells [34]. In surveys after natural disasters such as earthquakes and hurricanes, associations between disaster-related stress and decrease of immune functions, especially suppression of NK cell activity, have been reported [34–38]. A survey of victims of the 1995 Southern Hyogo Prefecture Earthquake in Japan revealed that post-traumatic stress disorder symptoms and poor lifestyle are associated with low NK cell activity [38].

Reconstruction of the healthcare system after the Great East Japan Earthquake is significant; however, the progress is uneven among the affected areas [39]. Design and construction of a comprehensive medical system, including lifestyle habit improvements, mental health care, and medical screening after a disaster, are essential for preventing health damage to women. During the sub-acute and chronic phases following the Great East Japan Earthquake, there were significant needs for medical and public health assistance that included infectious disease control and mental health care at evacuation facilities [40]. The provision of health and health facilities is a priority issue in the international guidelines on disaster reduction actions for the 15 years to 2030 adopted by the 3rd United Nations World Conference on Disaster Reduction (WCDRR, 2015) [41]. The present study suggests the importance of prompt reconstruction of healthcare systems and the inclusion of cancer screening in these systems to maintain women's health.

## Supporting information

**S1 Table. Areas where cervical cancer screening is covered by the Miyagi Cancer Association.**
(XLSX)

**S1 Fig. Change in the cervical cancer screening rate (CCS-R) in 23 areas that were screened at the hospital of a medical center in Miyagi Prefecture.** The bar graph shows the CCS-R (left vertical axis; unit, %), and the line graph shows the population (right vertical axis; unit, number of people).
(PPTX)

## Acknowledgments

The authors would like to acknowledge all of the staff at the Miyagi Cancer Society (Sendai, Japan).

## Author Contributions

**Conceptualization:** Yasuhiro Miki, Kiyoshi Ito.

**Data curation:** Yasuhiro Miki, Toru Tase, Hideki Tokunaga.

**Formal analysis:** Yasuhiro Miki, Toru Tase, Hideki Tokunaga.

**Investigation:** Yasuhiro Miki, Hideki Tokunaga, Nobuo Yaegashi.

**Supervision:** Toru Tase, Nobuo Yaegashi, Kiyoshi Ito.

**Writing – original draft:** Yasuhiro Miki.

**Writing – review & editing:** Nobuo Yaegashi, Kiyoshi Ito.

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
