## [Decision Letter · Decision Letter 0]

17 Dec 2019

PONE-D-19-25407

Cervical cancer screening rates before and after the Great East Japan Earthquake in Miyagi prefecture, Japan

PLOS ONE

Dear Dr. Ito,

Thank you for submitting your manuscript to PLOS ONE. After careful consideration, we feel that it has merit but does not fully meet PLOS ONE’s publication criteria as it currently stands. Therefore, we invite you to submit a revised version of the manuscript that addresses the points raised during the review process.

The reviewers addressed several major and minor concerns about your manuscript. Please revise your manuscript carefully.

We would appreciate receiving your revised manuscript by Jan 31 2020 11:59PM. To enhance the reproducibility of your results, we recommend that if applicable you deposit your laboratory protocols in protocols.io, where a protocol can be assigned its own identifier (DOI) such that it can be cited independently in the future. For instructions see: http://journals.plos.org/plosone/s/submission-guidelines#loc-laboratory-protocols

We look forward to receiving your revised manuscript.

Kind regards,

Kenji Hashimoto, PhD

Academic Editor

PLOS ONE

Journal Requirements:

**When submitting your revision, we need you to address these additional requirements:**

**Please ensure that your manuscript meets PLOS ONE's style requirements, including those for file naming. The PLOS ONE style templates can be found at http://www.plosone.org/attachments/PLOSOne_formatting_sample_main_body.pdf and http://www.plosone.org/attachments/PLOSOne_formatting_sample_title_authors_affiliations.pdf**PLOS requires an ORCID iD for the corresponding author in Editorial Manager on papers submitted after December 6th, 2016. Please ensure that you have an ORCID iD and that it is validated in Editorial Manager. To do this, go to ‘Update my Information’ (in the upper left-hand corner of the main menu), and click on the Fetch/Validate link next to the ORCID field. This will take you to the ORCID site and allow you to create a new iD or authenticate a pre-existing iD in Editorial Manager. Please see the following video for instructions on linking an ORCID iD to your Editorial Manager account: https://www.youtube.com/watch?v=_xcclfuvtxQ We note that [Figure 1A and 1B] in your submission contain [map/satellite] images which may be copyrighted. All PLOS content is published under the Creative Commons Attribution License (CC BY 4.0), which means that the manuscript, images, and Supporting Information files will be freely available online, and any third party is permitted to access, download, copy, distribute, and use these materials in any way, even commercially, with proper attribution. For these reasons, we cannot publish previously copyrighted maps or satellite images created using proprietary data, such as Google software (Google Maps, Street View, and Earth). For more information, see our copyright guidelines: http://journals.plos.org/plosone/s/licenses-and-copyright.We require you to either (1) present written permission from the copyright holder to publish these figures specifically under the CC BY 4.0 license, or (2) remove the figures from your submission:a.)You may seek permission from the original copyright holder of Figure 1A and 1B to publish the content specifically under the CC BY 4.0 license. We recommend that you contact the original copyright holder with the Content Permission Form (http://journals.plos.org/plosone/s/file?id=7c09/content-permission-form.pdf) and the following text:“I request permission for the open-access journal PLOS ONE to publish XXX under the Creative Commons Attribution License (CCAL) CC BY 4.0 (http://creativecommons.org/licenses/by/4.0/). Please be aware that this license allows unrestricted use and distribution, even commercially, by third parties. Please reply and provide explicit written permission to publish XXX under a CC BY license and complete the attached form.”Please upload the completed Content Permission Form or other proof of granted permissions as an "Other" file with your submission.In the figure caption of the copyrighted figure, please include the following text: “Reprinted from [ref] under a CC BY license, with permission from [name of publisher], original copyright [original copyright year].”b.) If you are unable to obtain permission from the original copyright holder to publish these figures under the CC BY 4.0 license or if the copyright holder’s requirements are incompatible with the CC BY 4.0 license, please either i) remove the figure or ii) supply a replacement figure that complies with the CC BY 4.0 license. Please check copyright information on all replacement figures and update the figure caption with source information. If applicable, please specify in the figure caption text when a figure is similar but not identical to the original image and is therefore for illustrative purposes only.The following resources for replacing copyrighted map figures may be helpful:USGS National Map Viewer (public domain): http://viewer.nationalmap.gov/viewer/The Gateway to Astronaut Photography of Earth (public domain): http://eol.jsc.nasa.gov/sseop/clickmap/Maps at the CIA (public domain): https://www.cia.gov/library/publications/the-world-factbook/index.html and https://www.cia.gov/library/publications/cia-maps-publications/index.htmlNASA Earth Observatory (public domain): http://earthobservatory.nasa.gov/Landsat: http://landsat.visibleearth.nasa.gov/USGS EROS (Earth Resources Observatory and Science (EROS) Center) (public domain): http://eros.usgs.gov/#Natural Earth (public domain): http://www.naturalearthdata.com/

Reviewers' comments:

Reviewer's Responses to Questions

**Comments to the Author**

1. Is the manuscript technically sound, and do the data support the conclusions?

Reviewer #1: Partly

Reviewer #2: No

Reviewer #3: Yes

2. Has the statistical analysis been performed appropriately and rigorously? 

Reviewer #1: I Don't Know

Reviewer #2: No

Reviewer #3: Yes

3. Have the authors made all data underlying the findings in their manuscript fully available?

Reviewer #1: Yes

Reviewer #2: Yes

Reviewer #3: Yes

4. Is the manuscript presented in an intelligible fashion and written in standard English?

Reviewer #1: Yes

Reviewer #2: No

Reviewer #3: Yes

5. Review Comments to the Author

Reviewer #1: PLOS ONE Sept 2019

Maybe have an editor copy edit for grammar.

Abstract: say earthquake was a magnitude 9.0 to show the seriousness of the disaster.

Intro: authors say screening rates are low in Japan compared to other countries. What are the prevalence rates of cervical cancer in Japan compared to other countries? What about HPV vaccine rates?

Why was it called the 3.11 earthquake?

What is the evidence to support this line: However, the people affected by the disaster are

61 still unstable, and cannot afford to take care of their health.

Why study cervical cancer and not the other women’s health conditions? Not well justified

Table 1 is not useful to a reader unfamiliar with the region. A map would be more useful.

Methods: Is the Miyagi Cancer Society a reputable resource? How do they get their data? Is it a government agency?

Results: Why only mobile van data included? Seems like a significant limitation. Where do most people get cervical cancer screenings in this region?

Better visualizations would make the data more meaningful (a map rather than a long table, perhaps). AH OK, I see maps at the end. I like the figures. But better to label the map with the region name and the bar charts with region names. The “T-3” labels are hard to follow.

Discussion: what were the results from the pregnancy studies? I still don’t know why cervical cancer was the focus here. No solid health or financial justification has been provided. OK page 19-21 are good. Maybe more of this justification or background should be in the introduction.

What was the population decline?

Overall it’s a decent paper. Though I am still unclear if the magnitude of change in screening is clinically meaningful. Can that point be proven? For example, saying that in Katrina the mean time to diagnosis was 7 years compared to 4 before Katrina – that is clinically meaningful. Could something similar be said here?

Reviewer #2: It may be more universally understandable if the magnitude of the earthquake is given in the abstract, i.e., after the Great East Japan Earthquake to put in the parentheses (magnitude 11.0); if the authors decide to leave 3.11 earthquake in the parentheses, they should put it in quotations "3.11" or indicate in some other way that this is a colloquial, synonymous term. The authors do explain this in line 53, however (but the abstract is often the first paragraph read in a manuscript). Elsewhere in the manuscript (line 53, 59), it may be more clear if the authors refer to the earthquake as the Great East Japan Earthquake instead of the 3.11 earthquake.

In the abstract, the sentence "It is possible that a decrease in CCS-R will lead to the detection in the advanced stages of cancer." may be incorrect? Did the authors want to suggest that a decrease in CCS-R may lead to less detection of advanced stages of cancer?

In the first sentence of the introduction (line 36), please state why this recommendation is important (otherwise this sentence seems abrupt as an opening to this manuscript).

The introduction does not flow well in terms of the verbage used. Lines 41-45 should flow and lead into each other.

Lines 55-57 use past and present tense terms. Instead of "may need 10 years for reconstruction" perhaps the authors may consider using "the tsunami-hit area was predicted to necessitate 10 years for reconstruction."

Please give a time course in lines 57-61. How long after the earthquake did the mobile van service resume screening?

Line 62: please cite this publication within the manuscript using a standardized method such as MLA format (i.e., "In a study describing survey results regarding reproductive health after the 2008 Wenchuan earthquake in China, Liu et al. reported......")

Lines 69-71 is too nebulous and general of a statement to make. The authors should add in that the failure of health management in the specific context of natural disasters (or whatever they feel it is specifically pertaining to).

Line 72: "Japan fiscal year (FY; April 1 to March 31) 2009–FY2016" should be simplified to set years (i.e., 2009-2016).

Table 1 is too complex and slightly unnecessary. Perhaps a small color coded index included as a legend in Figure 1 with the names of the prefectures would suffice (with consideration to take out Table 1 entirely; the prefectures can remain in Table 2 since it also provides pertinent information).

Line 93: instead of "people," should this be "females"? Why is the CCS-R pertaining to those 20 years old and over - if this is a screening protocol in Japan then please state that here.

It would be useful to get an idea about the numbers affected since -3.0% etc...(e.g., line 113, line 121) may be difficult to interpret meaningfully to the reader. The authors should consider detailing the population numbers in the text and delineating the axes on the graphs in all of the figures themselves (including the supplementary ones) for whatever they represent (i.e., year, population in thousands etc...) instead of in the figure legend.

The authors may consider not stating: "no significant difference was found" in line 170 since the p value was 0.6303. Likewise, in line 180 it is stated that there was a significant decrease in the coastal area compared with that in other areas however this is difficult to interpret without a regression analysis (and consideration of determining a p value).

Lines 196-200 are unclear in their message?

Lines 204-206: even though the authors could not identify causes, are there any that could be hypothesized based on data from previous similar studies?

I was not able to easily find the information for reference #23. Is New Orlando, USA a city that was described in this study?

Line 218: what stage is being referred to? Lines 219-221 should be considered being combined.

Why is this important for clinical outcomes? Is there any data regarding how many cervical cancers were missed as a result of delayed screening (either from the authors data set or from any of the references)?

Lines 235-236 contain repeated information which was just mentioned in lines 232-234.

Lines 239-242 need to be written with a more clear introduction or transition. Lines 241-242 are unclear.

Consider writing a separate conclusion since lines 244-248 do not transition in a clear way since the discussion prior to it pertains to smoking a mental stress.

The conclusive remark (line 246-248) is meaningful however there is lack of data to support this claim. The authors should consider including evidence from the literature supporting why design and construction of a comprehensive medical system etc... would be beneficial.

Reviewer #3: The study utilized appropriate statistics in analyzing data collected from relevant sources. The conclusions drawn and recommendations made were based on the results of the study. However, the following observations and comments addressed.

1. The cervical cancer screening rate for Japan should be stated for adequate comparison with other countries (lines 40-41).

2. Recast the sentence on lines 50 - 51 removing 'screening for cervical cancer screening' so the sentence reads 'All mobile van screenings are population-based'.

3. Replace 'as per' with a standard English word or phrase (line 65).

4. Replace 'i.e' with the appropriate words (line 97).

5. Lines 196 - 197: Report in the past.

References.

6. Cross-check if the following articles are single-paged articles: reference nos [3], [14], [21], [39]; and provide the complete pages where missing.

7. Provide the year of publication for no [5], and page numbers for no [27]. Also check the correctness of the page number for reference no [38].

6. PLOS authors have the option to publish the peer review history of their article (what does this mean?). If published, this will include your full peer review and any attached files.

Reviewer #1: No

Reviewer #2: No

Reviewer #3: Yes: Golda Ekenedo Ph.D.

---

## [Author Response · Author response to Decision Letter 0]

30 Jan 2020

To Reviewer #1: 

We appreciate your comments regarding our manuscript entitled “Cervical cancer screening rates before and after the Great East Japan Earthquake in Miyagi prefecture, Japan” (manuscript ID: PONE-D-19-25407). We have revised the manuscript according to your comments and provided detailed responses to the specific comments below.

Maybe have an editor copy edit for grammar.

This manuscript has been strictly edited.

Abstract: say earthquake was a magnitude 9.0 to show the seriousness of the disaster.

We have added the above to the abstract accordingly (lines 22-23).

Intro: authors say screening rates are low in Japan compared to other countries. What are the prevalence rates of cervical cancer in Japan compared to other countries? What about HPV vaccine rates?

We appreciate your comment. We believe it is important to provide information on cervical cancer prevalence and vaccination rates in Japan. We have summarized that information in the Introduction section (lines 36-44; 46-47; 49-50). 

Why was it called the 3.11 earthquake?

The abbreviation "3.11" has been used mainly by Japanese mass media since the earthquake occurred on March 11, 2011. We described the above in the Introduction section (lines 57-59). In addition, “3.11” is widely recognized in Japan; however, there may be many readers who are not familiar with this term. Therefore, we have not used “3.11” as an abbreviation in the manuscript. 

What is the evidence to support this line: However, the people affected by the disaster are 61 still unstable, and cannot afford to take care of their health.

Several studies have reported that the mental and financial instability of survivors creates “patients’ delay” (lines 70-71). We have added references to clarify this and described it in the next paragraph in the revised manuscript (lines 72-82). 

Why study cervical cancer and not the other women’s health conditions? Not well justified

We have summarized this information in the Introduction and have described it in the Discussion section. We clarified why we targeted a disaster and its impact on cervical cancer screening in this research (lines 72-86).

Table 1 is not useful to a reader unfamiliar with the region. A map would be more useful.

The code numbers in Table 1, which is S1 Table in the revised version, match those in Figure 1B in order to distinguish the areas on the map. It is clearly described in each legend in the S1 Table and Figure 1B in the revised manuscript. 

Methods: Is the Miyagi Cancer Society a reputable resource? How do they get their data? Is it a government agency?

The Miyagi Cancer Society was the first established organization in Japan for cancer since 1958. In 2012, the society was recognized by the administrative agency as a public interest incorporated foundation and has undertaken a cancer screening and cancer registration business in Miyagi Prefecture, Japan. The Annual Report is also compiled as one of its functions. We have summarized these details in the Materials and Methods (lines 100-103). 

Results: Why only mobile van data included? Seems like a significant limitation. Where do most people get cervical cancer screenings in this region?

Medical exam statics at hospitals or the medical center in Miyagi Prefecture include a large number of occasional outpatients, who complain of symptoms such as irregular bleeding. On the other hand, women who have undergone cervical cancer screening in a mobile van were all periodic health care check-ups. In addition, in these regions, cervical cancer screening is covered by the mobile van. We added above sentences in the Materials and Methods and Results (lines 117-122). 

Better visualizations would make the data more meaningful (a map rather than a long table, perhaps). AH OK, I see maps at the end. I like the figures. But better to label the map with the region name and the bar charts with region names. The “T-3” labels are hard to follow.

We agree with your comment. We revised Figure 1B accordingly. 

Discussion: what were the results from the pregnancy studies? I still don’t know why cervical cancer was the focus here. No solid health or financial justification has been provided. OK page 19-21 are good. Maybe more of this justification or background should be in the introduction.

We agree with your suggestion. We have included a summary of this discussion in the introduction and have clarified our motivations for studying a disaster and its impact on cervical cancer screening (lines 72-86). 

What was the population decline?

It is well known that the tsunami resulted in severe human and residential damage as a result of the Great East Japan Earthquake. Therefore, coastal populations are considered to have decreased both due to death as well as migration as a consequence of the tsunami. We added this sentence in the Discussion section (lines 227-230). 

Overall it’s a decent paper. Though I am still unclear if the magnitude of change in screening is clinically meaningful. Can that point be proven? For example, saying that in Katrina the mean time to diagnosis was 7 years compared to 4 before Katrina – that is clinically meaningful. Could something similar be said here?

We appreciate your kind comments regarding our research. Compared to the USA CCS-R, the Japan CCS-R is too low. Therefore, it would be difficult to perform the same analysis as that in US research due to the small number of comparative patients. Otherwise, in Japan, cervical cancer screening is expected to reduce cervical cancer deaths (recommendation grade B), with evidence levels of 2++ for conventional and 2+ for liquid-based cytology. There are limitations to conducting a further detailed analysis in the areas examined in this study; however, a reduction in CCS-Rs suggests that women’s health will be significantly affected. We have clarified this aspect in the Discussion section (lines 274-286). 

 

To Reviewer #2

We appreciate your comments regarding our manuscript entitled “Cervical cancer screening rates before and after the Great East Japan Earthquake in Miyagi prefecture, Japan” (manuscript ID: PONE-D-19-25407). We have revised the manuscript according to your comments and provided detailed responses to specific comments below.

It may be more universally understandable if the magnitude of the earthquake is given in the abstract, i.e., after the Great East Japan Earthquake to put in the parentheses (magnitude 11.0); if the authors decide to leave 3.11 earthquake in the parentheses, they should put it in quotations "3.11" or indicate in some other way that this is a colloquial, synonymous term. The authors do explain this in line 53, however (but the abstract is often the first paragraph read in a manuscript). Elsewhere in the manuscript (line 53, 59), it may be more clear if the authors refer to the earthquake as the Great East Japan Earthquake instead of the 3.11 earthquake.

We agree with your suggestions. In the abstract, “3.11” has not been used as an abbreviation for the Great East Japan Earthquake. In the text, “3.11” was only mentioned as being used mainly by the Japanese mass media and not used as an abbreviation elsewhere (lines 57-59). 

In the abstract, the sentence "It is possible that a decrease in CCS-R will lead to the detection in the advanced stages of cancer." may be incorrect? Did the authors want to suggest that a decrease in CCS-R may lead to less detection of advanced stages of cancer?

We apologize for any inaccuracies. As you pointed out, a decrease in the CCS-R may lead to low detection of advanced stages of cancer. We have revised the abstract accordingly (lines 31-32). 

In the first sentence of the introduction (line 36), please state why this recommendation is important (otherwise this sentence seems abrupt as an opening to this manuscript).

We agree with your comments. It is important to consider that in the results of this study, cervical cancer screening is recommended based on the evidence from Japan. Therefore, we moved this sentence from the Introduction to the Discussion section in the revised manuscript. 

The introduction does not flow well in terms of the verbage used. Lines 41-45 should flow and lead into each other.

We agree with your comments. We have revised this series of sentences in the revised manuscript (lines 36-55). 

Lines 55-57 use past and present tense terms. Instead of "may need 10 years for reconstruction" perhaps the authors may consider using "the tsunami-hit area was predicted to necessitate 10 years for reconstruction."

We agree with your comments. We have revised this sentence accordingly (lines 62-63).

Please give a time course in lines 57-61. How long after the earthquake did the mobile van service resume screening?

We agree with your comments. In many areas, cervical cancer screening was resumed in April of the year of the disaster . However, in coastal areas (L-2, L-3, T-3, U-1b in Fig. 1), the re-initiation of screenings was delayed from July to December of that same year . The screening was resumed only in Oshika (L-7 in Fig. 1) in February following the earthquake. We added this information to the Introduction section (lines 66-70). 

Line 62: please cite this publication within the manuscript using a standardized method such as MLA format (i.e., "In a study describing survey results regarding reproductive health after the 2008 Wenchuan earthquake in China, Liu et al. reported......")

We agree with your comments. In the revised manuscript, we modified the Introduction section to include this sentence (lines 72-86). 

Lines 69-71 is too nebulous and general of a statement to make. The authors should add in that the failure of health management in the specific context of natural disasters (or whatever they feel it is specifically pertaining to).

We agree with your comments. In the revised manuscript, we modified the Introduction section accordingly (lines 72-86). 

Line 72: "Japan fiscal year (FY; April 1 to March 31) 2009–FY2016" should be simplified to set years (i.e., 2009-2016).

We agree with your comments. We have revised the manuscript accordingly (line 87).

Table 1 is too complex and slightly unnecessary. Perhaps a small color coded index included as a legend in Figure 1 with the names of the prefectures would suffice (with consideration to take out Table 1 entirely; the prefectures can remain in Table 2 since it also provides pertinent information).

We agree that Table 1 may be unnecessary for many readers. However, it may be helpful for readers who know Japan in order to identify a location. We have left it as a supplemental table. In addition, we have revised Figure 1 to make it easier to understand. 

Line 93: instead of "people," should this be "females"? Why is the CCS-R pertaining to those 20 years old and over - if this is a screening protocol in Japan then please state that here.

We appreciate your polite comment. Line 93 "people" has been corrected to "women". 

The incidence of cervical cancer has increased in Japanese women in their mid-20s. Therefore, Japanese cervical cancer screening guidelines recommend to begin screening at the age of 20 years. We described this in the Materials and Methods section (lines 105-107).

It would be useful to get an idea about the numbers affected since -3.0% etc...(e.g., line 113, line 121) may be difficult to interpret meaningfully to the reader. The authors should consider detailing the population numbers in the text and delineating the axes on the graphs in all of the figures themselves (including the supplementary ones) for whatever they represent (i.e., year, population in thousands etc...) instead of in the figure legend.

We agree with your comments. We have added new graphs in Figures 2 and 3 to make it easier to understand the decrease in cervical cancer screening rates. We also revised the graphs accordingly. 

The authors may consider not stating: "no significant difference was found" in line 170 since the p value was 0.6303. Likewise, in line 180 it is stated that there was a significant decrease in the coastal area compared with that in other areas however this is difficult to interpret without a regression analysis (and consideration of determining a p value).

We apologize for failing to explain Figure 5. We divided the regions into coastal and non-coastal areas and compared them using the Mann–Whitney U test. The p-value shows the results. We have added the legend for Figure 5 (lines 211-215)..

Lines 196-200 are unclear in their message?

In this article, it was suggested that the victims may be motivated to manage their health if they realize that there is recovery from disaster. However, there is no evidence or reference to support this suggestion as you point out. I agree that this is unclear. Therefore, we have deleted this text from the revised manuscript. 

Lines 204-206: even though the authors could not identify causes, are there any that could be hypothesized based on data from previous similar studies?

We agree with your comments. In an evidence review by CPSTF (USA), small media sources such as newspapers and educational videos as well as solicitation by telephone or letter are recommended as interventions to increase the CCS-R (https://www.thecommunityguide.org/content/task-force-findings-cancer-prevention-and-control; access 2020.1.17). In Japan, it was also reported that local government initiatives and education contributed to increasing the screening rate of young women. However, a detailed investigation is required in areas where these screening rates have declined and awareness attempts by local government and other organizations may have been delayed. We have added this to the Discussion section (lines 248-257). 

I was not able to easily find the information for reference #23. Is New Orlando, USA a 

city that was described in this study?

Ref.#23 is an abstract of the 2001 Western Association of Gynecologic Oncologists (WAGO) Annual Meeting published in Gynecologic Oncology. We noted this in the manuscript that this reference is a conference abstract. The DOI number is also added to link to this abstract (lines 78-80). 

Line 218: what stage is being referred to? Lines 219-221 should be considered being combined.

The details of the stage data are described in the Discussion section. We also revised the manuscript as you pointed out (lines 270-273).

Why is this important for clinical outcomes? Is there any data regarding how many cervical cancers were missed as a result of delayed screening (either from the authors data set or from any of the references)?

The five-year overall survival rates for cervical cancer stages I and II are known to be approximately 90% and 75%, respectively. Early detection by periodic screening is critical in order to prevent death from cervical cancer. If regular screening is skipped as a result of the earthquake, it may be detected later as advanced cancer. This condition was found in the area that was impacted by Hurricane Katrina.We added this in the Discussion section (lines 273-282).

Lines 235-236 contain repeated information which was just mentioned in lines 232-234.

We deleted this sentence in the revised manuscript.

Lines 239-242 need to be written with a more clear introduction or transition. Lines 241-242 are unclear.

It is known that smoking increases the risk of HPV infection. In this sentence, it suggests the number of smokers is increasing in many affected areas, implying that cervical cancer should be considered in these areas. However, there is no direct evidence of smoking rates in coastal and non-coastal areas surveyed in this research. Therefore, we removed this information from the revised manuscript.

Consider writing a separate conclusion since lines 244-248 do not transition in a clear way since the discussion prior to it pertains to smoking a mental stress.

We removed the description of mental stress and smoking and added a separate conclusion in the revised manuscript.

The conclusive remark (line 246-248) is meaningful however there is lack of data to support this claim. The authors should consider including evidence from the literature supporting why design and construction of a comprehensive medical system etc... would be beneficial.

During the sub-acute and chronic phases following the Great East Japan Earthquake, there were significant needs for medical and public health assistance that included infectious disease control and mental health care at evacuation facilities. The provision of health and health facilities is a priority issue in the international guidelines on disaster reduction actions for the 15 years to 2030 adopted by the 3rd United Nations World Conference on Disaster Reduction (WCDRR, 2015). The present study suggests the importance of prompt reconstruction of healthcare systems and the inclusion of cancer screening in these systems to maintain women's health. We have described this in the Discussion section as a conclusion (lines 306-314). 

 

To Reviewer #3: 

We appreciate your comments regarding our manuscript entitled “Cervical cancer screening rates before and after the Great East Japan Earthquake in Miyagi prefecture, Japan” (manuscript ID: PONE-D-19-25407). We have revised the manuscript according to your comments and provided detailed responses to specific comments below.

The study utilized appropriate statistics in analyzing data collected from relevant sources. The conclusions drawn and recommendations made were based on the results of the study. However, the following observations and comments addressed.

1. The cervical cancer screening rate for Japan should be stated for adequate comparison with other countries (lines 40-41).

We have added cervical cancer screening rates for Japan (33.7%; age group: 20-69 years) and Miyagi Prefecture (42.1%; age group: 20-69 years) (lines 46-50).

2. Recast the sentence on lines 50 - 51 removing 'screening for cervical cancer screening' so the sentence reads 'All mobile van screenings are population-based'.

We appreciate your comments. We have revised this sentence accordingly.

3. Replace 'as per' with a standard English word or phrase (line 65).

This phrase was considered meaningless and has been removed from the revised manuscript.

4. Replace 'i.e' with the appropriate words (line 97).

This word has been corrected to "e.g." (line 113).

5. Lines 196 - 197: Report in the past.

We appreciate your comment. This sentence was deleted due to the other reviewers' suggestions. 

References.

6. Cross-check if the following articles are single-paged articles: reference nos [3], [14], [21], [39]; and provide the complete pages where missing.

These references are single-paged online journals.

7. Provide the year of publication for no [5], and page numbers for no [27]. Also check the correctness of the page number for reference no [38].

Reference No. [5] was 2016 public data of National Cancer Center in Japan. Nos. [27] and [28] are also online journals, each having a single number "4" and "e018943".

---

## [Decision Letter · Decision Letter 1]

11 Feb 2020

PONE-D-19-25407R1

Cervical cancer screening rates before and after the Great East Japan Earthquake in Miyagi Prefecture, Japan

PLOS ONE

Dear Dr. Ito,

Thank you for submitting your manuscript to PLOS ONE. After careful consideration, we feel that it has merit but does not fully meet PLOS ONE’s publication criteria as it currently stands. Therefore, we invite you to submit a revised version of the manuscript that addresses the points raised during the review process.

The reviewer #3 addressed some minor comments about your revised manuscript. Please check the manuscript carefully.

We would appreciate receiving your revised manuscript by Mar 27 2020 11:59PM. To enhance the reproducibility of your results, we recommend that if applicable you deposit your laboratory protocols in protocols.io, where a protocol can be assigned its own identifier (DOI) such that it can be cited independently in the future. For instructions see: http://journals.plos.org/plosone/s/submission-guidelines#loc-laboratory-protocols

We look forward to receiving your revised manuscript.

Kind regards,

Kenji Hashimoto, PhD

Academic Editor

PLOS ONE

Reviewers' comments:

Reviewer's Responses to Questions

**Comments to the Author**

1. If the authors have adequately addressed your comments raised in a previous round of review and you feel that this manuscript is now acceptable for publication, you may indicate that here to bypass the “Comments to the Author” section, enter your conflict of interest statement in the “Confidential to Editor” section, and submit your "Accept" recommendation.

Reviewer #2: All comments have been addressed

Reviewer #3: (No Response)

2. Is the manuscript technically sound, and do the data support the conclusions?

Reviewer #2: Partly

Reviewer #3: Yes

3. Has the statistical analysis been performed appropriately and rigorously? 

Reviewer #2: Yes

Reviewer #3: Yes

4. Have the authors made all data underlying the findings in their manuscript fully available?

Reviewer #2: Yes

Reviewer #3: Yes

5. Is the manuscript presented in an intelligible fashion and written in standard English?

Reviewer #2: No

Reviewer #3: Yes

6. Review Comments to the Author

Reviewer #2: (No Response)

Reviewer #3: Authors should make the following corrections:

Lines 39 - 40; the data source should be appropriately referenced by including it in the reference list and represented with the reference number in block parenthesis in the text.

Line 113; the appropriate phrase for i.e, is 'that is', not e.g.

7. PLOS authors have the option to publish the peer review history of their article (what does this mean?). If published, this will include your full peer review and any attached files.

Reviewer #2: No

Reviewer #3: Yes: Golda O. Ekenedo

---

## [Author Response · Author response to Decision Letter 1]

17 Feb 2020

To Reviewer #3: 

We appreciate your comments regarding our manuscript entitled “Cervical cancer screening rates before and after the Great East Japan Earthquake in Miyagi prefecture, Japan” (manuscript ID: PONE-D-19-25407R1). We have revised the manuscript according to your comments and provided detailed responses to specific comments below.

Authors should make the following corrections:

Lines 39 - 40; the data source should be appropriately referenced by including it in the reference list and represented with the reference number in block parenthesis in the text.

We agree with your comments. We have revised manuscript accordingly (line 39 and reference #1).

Line 113; the appropriate phrase for i.e, is 'that is', not e.g.

We appreciate your comments. We have corrected manuscript accordingly (line 112).

---

## [Editor Report · Decision Letter 2]

19 Feb 2020

Cervical cancer screening rates before and after the Great East Japan Earthquake in Miyagi Prefecture, Japan

PONE-D-19-25407R2

Dear Dr. Ito,

We are pleased to inform you that your manuscript has been judged scientifically suitable for publication and will be formally accepted for publication once it complies with all outstanding technical requirements.

With kind regards,

Kenji Hashimoto, PhD

Section Editor

PLOS ONE
---

## [Editor Report · Acceptance letter]

25 Feb 2020

PONE-D-19-25407R2 

Cervical cancer screening rates before and after the Great East Japan Earthquake in the Miyagi Prefecture, Japan 

Dear Dr. Ito:

I am pleased to inform you that your manuscript has been deemed suitable for publication in PLOS ONE. Congratulations! Your manuscript is now with our production department. 

With kind regards,

on behalf of

Prof. Kenji Hashimoto 

Section Editor

PLOS ONE